# Soluble Neuropilin-1 as a Marker for Distinguishing Bacterial and Viral Sepsis in Critically Ill Patients—A Prospective, Multicenter, Observational Study

**DOI:** 10.3390/v17070997

**Published:** 2025-07-16

**Authors:** Fabian Perschinka, Georg Franz Lehner, Timo Mayerhöfer, Frank Hartig, Birgit Zassler, Johannes Bösch, Dietmar Fries, Romuald Bellmann, Michael Joannidis

**Affiliations:** 1Division of Intensive Care and Emergency Medicine, Department of Internal Medicine, Medical University Innsbruck, 6020 Innsbruck, Austria; 2Department of General and Surgical Intensive Care Medicine, Medical University Innsbruck, 6020 Innsbruck, Austria

**Keywords:** critically ill, sepsis, soluble Neuropilin-1, bacterial infection, viral infection, inflammation

## Abstract

Sepsis causes millions of deaths each year. Rapid, targeted therapy can reduce mortality rates. Both bacterial and viral pathogens can trigger sepsis, but the utility of commonly used inflammatory markers for differentiation remains controversial. Moreover, little is known about the time courses of alternative inflammatory parameters. The aim of this prospective, two-center observational study was to investigate the differences in the course of soluble Neuropilin-1 (sNRP-1) levels between bacterial and viral sepsis over a 7-day period. To be included, adult patients had to meet the SEPSIS-3 criteria and be diagnosed with either a bacterial or viral pathogen. Immunosuppressed patients were excluded. While IL-6, PCT, and CRP levels decreased consistently over time, sNRP-1 levels remained elevated in the bacterial group throughout the entire ICU stay. PCT (*p* < 0.001) and CRP (*p* = 0.016) levels were significantly associated with the course of sNRP-1. The AUC of sNRP-1 was 0.777 for discriminating between bacterial and viral infections on day 1. sNRP-1 remained stable and significantly higher in bacterial than in viral infections. Furthermore, the AUC values for discrimination ranged from acceptable to good, depending on the day of the ICU stay. sNRP-1 may serve as a potential tool to differentiate between bacterial and viral pathogens in sepsis.

## 1. Introduction

Sepsis is defined as life-threatening organ dysfunction caused by a dysregulated host response to infection and accounts for millions of deaths worldwide each year. The presence of organ dysfunction and a dysregulated host response distinguishes sepsis from uncomplicated infection [1].

Bacterial pathogens are responsible for the majority of sepsis cases; however, fungal and viral pathogens can also cause infections that fulfill the SEPSIS-3 criteria [2,3]. Although blood cultures are negative in up to 42% of cases [3,4], suggesting a non-bacterial cause, non-bacterial and non-fungal organisms are cultured in fewer than 5% of cases [5]. The most common viruses requiring intensive care unit (ICU) admission were influenza, respiratory syncytial virus, and severe acute respiratory syndrome coronavirus 2 (SARS-CoV-2) [6,7].

The utility of commonly used inflammatory markers (interleukin-6 [IL-6], C-reactive protein [CRP], procalcitonin [PCT]) for differentiating between bacterial and viral pathogens remains controversial [8], and their levels are highly dynamic throughout the course of disease. However, little is known about the progression and variability of other immune-related parameters.

Since none of the established markers provides convincing accuracy on its own, we sought to investigate an alternative, not yet widely established parameter involved in the initiation of the immune response. Neuropilin-1 (NRP-1) is a transmembrane glycoprotein initially identified in the nervous system [9]. Beyond its neuronal role, NRP-1 is now recognized as a multifunctional coreceptor for a range of ligands, including semaphorin 3A, vascular endothelial growth factor (VEGF), and transforming growth factor-beta (TGF-β), implicating it in diverse biological processes [10,11,12].

In the immune system, NRP-1 is expressed on dendritic cells and a subset of regulatory T cells, where it plays a key role in mediating the interaction between dendritic cells and T cells [13,14]. Mechanistically, NRP-1 stabilizes interactions between dendritic cells and T cells [15]. This facilitates prolonged cell-to-cell contact necessary for efficient antigen presentation and T cell activation during the primary immune response [14]. Blocking NRP-1 via antibodies has been shown to reduce the proliferation of resting T cells [13,14], underscoring its functional importance in immune activation. NRP-1 potentially influences immune tolerance through the induction of T cell anergy and IL-10 secretion [16].

In addition to the membrane-bound form, soluble isoforms of NRP-1 (sNRP-1) exist [17]. sNRP-1 levels have been found to be elevated in critically ill trauma and surgical patients, correlating with markers of inflammation and coagulation, such as C-reactive protein, IL-6, and INR [18]. Although the exact biological role of sNRP-1 remains incompletely understood, these findings suggest a potential involvement in systemic inflammatory and coagulative responses.

The aim of this study was to investigate differences in the course of sNRP-1 levels between bacterial and viral sepsis in critically ill patients over a 7-day period.

## 2. Materials and Methods

### 2.1. Patients

Patients included in this prospective, observational study were recruited between 1 August 2021 and 1 April 2024 from two ICUs in Austria. All adult patients who met the sepsis criteria according to the “The Third International Consensus Definitions for Sepsis and Septic Shock (SEPSIS-3)” [1] were eligible for inclusion if they were expected to stay for at least 48 h in the ICU. The viruses had to be detected at least once by an antigen-based test or a PCR-based test to be considered in the viral sepsis group. Bacterial infections were diagnosed by at least one positive aerobic or anaerobic blood culture, a positive result from a PCR-based test for bacterial genome or bacterial antigen, or a strongly suspected infection with a radiologically proven focus. Studies have shown that computer tomography has good sensitivity and positive predictive value for detecting the site of infection [19,20]. Only patients who had either a viral or a bacterial infection at the time of inclusion were considered; those with concurrent viral and bacterial infections were excluded.

Patients were excluded in case of immunological insufficiency due to comorbidities (lymphoma and humoral/cellular deficiencies), chronic use of immunosuppressive medication, or if patients received chemotherapy or radiotherapy within the past 12 months. Additionally, patients with treatment limitations or restrictions on ongoing life support at ICU admission or a diagnosis of acute/chronic pancreatitis were excluded.

### 2.2. Blood Sampling and Cytokine Measurement

Blood samples were collected from existing arterial lines as part of routine blood sampling for 7 days. All sNRP-1 samples were centrifuged, and supernatant plasma was subsequently stored at −80 °C until analysis.

sNRP-1 was analyzed using the Total Soluble Neuropilin-1 ELISA Kit by Biomedica (Biomedica Medizinprodukte GmbH, Vienna, Austria). For the sNRP-1 measurement, a TECAN Infinite M200 Plate Reader (TECAN Trading AG, Männedorf, Switzerland) at 450 nm wavelength was used.

IL-6 and PCT were analyzed by an electrochemiluminescence immunoassay. CRP was measured by a particle-enhanced turbidimetric immunoassay.

### 2.3. Data Collection

Treatments were considered if they were performed for at least 2 h per day: invasive mechanical ventilation (IMV) comprises mechanical ventilation via endotracheal intubation or tracheostoma, while nasal high flow (NHF) and continuous positive airway pressure (CPAP) conducted by mask or helmet were classified as non-invasive ventilation (NIV). Acute kidney injury (AKI) was defined according to the Kidney Disease: Improving Global Outcomes (KDIGO) guidelines by increased serum creatinine or decreased urine output [21].

Bacterial and viral co-infections were taken into account from day 1 after study inclusion if the bacterial or viral pathogens were detected with one of the above-mentioned methods. Fungal co-infections were recorded from day 1 if they were detected by blood culture or analysis of bronchial secretion.

Data were collected until death or hospital discharge, whichever occurred earlier. Missing values were kept and not imputed.

All data were recorded in an electronic Case Report Form (eCRF) by REDCap electronic data capture, a web platform for managing databases and surveys created by Vanderbilt University [22,23].

### 2.4. Statistical Analysis

Continuous variables are presented as medians with interquartile ranges (IQRs), and categorical variables are shown as numbers with corresponding percentages. To analyze categorical variables, we used the χ^2^-test. We used the Shapiro–Wilk test to analyze the distribution of continuous variables. If a normal distribution was assumed, a *t*-test was performed. Otherwise, the Mann–Whitney U test was used for group comparisons.

Linear mixed models were used to estimate the effect of IL-6, PCT, or CRP on sNRP-1 levels over time. The area under the ROC curve (AUC) of inflammatory markers and sNRP-1 was calculated for differentiation between bacterial and viral pathogens.

In order to consider differences in the immune response of different sepsis foci, the aforementioned analyses were also performed in the subgroup of respiratory disease.

A *p*-value < 0.05 was considered statistically significant. All tests were conducted 2-sided.

For statistical analysis, the software SPSS (version: 29, IBM Corp. Armonk, NY, USA) was used.

## 3. Results

A total of 51 patients were included in this study, of whom 37 were diagnosed with bacterial infections and 14 with viral infections. In all patients classified as having bacterial sepsis, except one, a bacterial pathogen could be detected. All viral infections in the viral sepsis group were confirmed by PCR. The identified pathogens are listed in Appendix A. Both groups were comparable with respect to age and gender but showed differences in SAPS III as well as in SOFA scores. On the day of ICU admission, 94.4% in the bacterial sepsis group and 78.5% in the viral sepsis group required pharmacologic hemodynamic support. All viral sepsis cases were associated with a respiratory focus, whereas in the bacterial group, various foci were detected, with respiratory being the most frequent. Detailed baseline characteristics can be obtained in Table 1.

### 3.1. Treatment

In both groups, the majority of patients required invasive mechanical ventilation and vasopressor support during their ICU stay. The duration of invasive ventilation was slightly longer in the viral sepsis group compared to the bacterial group, although this difference was not statistically significant (11 days vs. 12 days; *p* = 0.699). While Dexamethasone was administered predominantly in the viral sepsis group, Hydrocortisone was more frequently used in bacterial sepsis patients. A detailed overview of therapeutic interventions and medication usage is provided in Appendix A.

### 3.2. Inflammatory Markers

The inflammatory markers showed significantly increased levels in the bacterial sepsis group compared to the viral sepsis group, particularly at ICU admission. Significantly higher median levels of IL-6 were observed in the group of bacterial infections in the first days of the ICU stay, with the highest median levels on the day of ICU admission (3900.5 ng/L [IQR: 984.0–44,140.0] vs. 145.0 ng/L [IQR: 68.4–190.0]; *p* < 0.001). No significant difference in IL-6 levels was seen between the groups from day 4 onwards. The initial high values in the bacterial sepsis group, which equalized with those in the viral sepsis group over time, were also observed in PCT and CRP. While IL-6 and CRP showed a rapid decrease, PCT levels declined more gradually throughout the ICU stay (Figure 1).

Details on co-infections diagnosed during the 7-day observation period, organized by day of detection, are summarized in Appendix A.

### 3.3. sNRP-1

Unlike IL-6, CRP, and PCT, sNRP-1 showed stable elevated levels in the bacterial sepsis group throughout the 7-day observation period. sNRP-1 levels in the bacterial sepsis group were significantly higher compared to the viral sepsis group, with the highest median levels of up to 3.38 nmol/L (IQR: 2.63–3.76; day 3) and the lowest median levels of 2.58 nmol/L (IQR: 1.92–3.22; day 7). In comparison, the viral sepsis group exhibited substantially lower levels, with the highest median concentration being 2.02 nmol/L (IQR: 1.26–2.75; day 1) (Table 2).

In the linear mixed models, a statistically significant association was found between both PCT (*p* = 0.029) as well as CRP (*p* = 0.016), and the course of sNRP-1. IL-6 (*p* = 0.305) did not show a comparable relationship. The discriminatory power of sNRP-1 for distinguishing bacterial from viral infections was acceptable, with an AUC of 0.777 on day 1. This discriminatory performance remained stable over the observation period, reaching an AUC of 0.846 on day 2 (Table 3). The AUC values of the other inflammatory markers are presented in the Appendix A. While IL-6 and CRP demonstrated lower discriminatory power, PCT showed comparable or slightly higher AUC values than sNRP-1.

### 3.4. Subgroup Analysis of Patients with Respiratory Focus

A subgroup analysis was performed on patients with a respiratory focus of infection. The baseline characteristics and treatments for this subgroup are provided in Appendix A.

In this subgroup analysis, sNRP-1 was also higher in the bacterial group compared to the viral sepsis patients. The median sNRP-1 levels in the bacterial group exceeded 2.50 nmol/L on all measured days, whereas the viral group predominantly exhibited levels below 2.00 nmol/L, except for a peak on day 1 (2.02 nmol/L) (Table 4). The courses of sNRP-1, IL-6, CRP, and PCT in this subgroup mirrored those observed in the overall cohort (Appendix A).

When restricted to patients with respiratory infections, the discriminatory ability of sNRP-1 to differentiate bacterial from viral sepsis on day 1 decreased slightly, with an AUC of 0.725 (Table 5).

## 4. Discussion

This is the first study to longitudinally evaluate the soluble Neuropilin-1 (sNRP-1) levels over a 7-day period in septic patients, comparing them to established inflammatory markers such as CRP, PCT, and IL-6. Our findings demonstrate distinct temporal patterns and concentrations depending on the underlying etiology of sepsis. Notably, the sNRP-1 levels remained significantly elevated in patients with bacterial sepsis compared to those with viral infections throughout the entire observation period.

NRP-1 plays a crucial role in the formation of the immunological synapse between dendritic cells and resting T cells—an interaction that marks an important step in the initiation of the primary immune response [13]. A positive correlation has been demonstrated between NRP-1 levels and the activation of Toll-like receptor 2 (TLR2) and Toll-like receptor 4 (TLR4) [24]. Both TLR2 and TLR4 are primarily involved in the detection of bacterial components [25]. These receptors initiate downstream MyD88-dependent signaling cascades, leading to NF-κB activation and subsequent proinflammatory cytokine release, including IL-6 [26]. Moreover, NRP-1 itself can enhance NF-κB activity by increasing its expression and DNA-binding capacity [27]. This mechanistic pathway supports our observation of higher sNRP-1 levels in bacterial sepsis and suggests that sNRP-1 may contribute to a heightened proinflammatory response. Consequently, sNRP-1 may serve as a valuable biomarker to distinguish between bacterial and viral etiologies in sepsis and septic shock.

The potential influence of co-infections on our findings appears limited, given that only a small number of bacterial co-infections were identified during the 7-day observation window. Due to the timing of culture positivity and the nature of the isolated organisms, many of these cases are more likely attributable to contamination rather than true secondary infections, particularly in the bacterial sepsis group.

Previous studies have demonstrated a positive correlation between the activation of Toll-like receptor 2 (TLR2) and Toll-like receptor 4 (TLR4) and the levels of NRP-1 [24]. TLR2 and TLR4 play key roles in the recognition of bacterial components [25], triggering the MyD88-dependent signaling pathway. This pathway culminates in the activation of nuclear factor kappa B (NF-κB), leading to the production of proinflammatory cytokines like IL-6 [26]. NRP-1 further amplifies this response by enhancing NF-κB activation, both through increased NF-κB expression and by strengthening its DNA-binding capacity [27]. Beyond its role in the initiation of inflammation, NRP-1 has also been implicated in immunosuppressive processes. In the context of sepsis, NRP-1 has been shown to stabilize regulatory T cells (TReg), promoting the release of immunomodulatory cytokines such as IL-4, IL-10, and TGF-β. These cytokines are key mediators in the development of sepsis-induced immunosuppression—a phenomenon associated with poor clinical outcomes and increased mortality [27,28]. Our finding that sNRP-1 levels remained elevated beyond the peak concentrations of traditional inflammatory markers (IL-6, CRP, and PCT) suggests that immunosuppressive mechanisms may persist even as proinflammatory signals decline.

Inflammatory markers such as IL-6, PCT, and CRP were markedly higher in bacterial sepsis at ICU admission, consistent with prior studies demonstrating a more robust cytokine response in bacterial versus viral infections [29]. A previous study reported AUC values of 0.89, 0.77, and 0.80 for IL-6, CRP, and PCT, respectively, when distinguishing septic patients from non-septic controls [30]. In our cohort, sNRP-1 demonstrated variable but generally acceptable to good discriminatory performance between the two patient populations that both met the SEPSIS-3 criteria, with the level of discrimination depending on the specific day of the ICU stay. This finding highlights its potential utility in supporting the differentiation between bacterial and viral sepsis—a distinction that remains clinically challenging.

We are not aware of any research on the (renal) elimination of sNRP-1. Its estimated molecular weight of approximately 90 kDa [31] suggests that renal clearance via glomerular filtration is unlikely. Additionally, specific transmembrane transporters in the tubule have not been described yet. One study suggests lysosomal degradation as the primary clearance mechanism for NRP-1 [32], indicating that organ dysfunction—renal or hepatic—is unlikely to account for the persistently elevated sNRP-1 levels observed in our cohort.

### Strengths and Limitations

This study has several notable strengths. Blood samples were consistently collected and processed across all patients during the observation period, ensuring methodological uniformity. All laboratory analyses were conducted by the same technician, minimizing inter-performer variability. However, the study is limited by the relatively small number of patients in the viral sepsis group and the predominance of SARS-CoV-2 infections within this subgroup. Additionally, data on the shedding mechanisms of transmembrane NRP-1 are lacking, limiting our understanding of sNRP-1 regulation.

While sNRP-1 showed differential expression patterns and potential diagnostic value, further research is needed to better understand its regulation and confirm its clinical utility in larger cohorts.

## 5. Conclusions

In contrast to conventional inflammatory markers, sNRP-1 remained significantly elevated in bacterial sepsis compared to viral sepsis over the entire observation period, without a significant decline. The diagnostic performance of sNRP-1, as indicated by the AUC values, varied between acceptable and good levels, depending on the day of the ICU stay. These findings suggest that sNRP-1 may serve as a promising biomarker for differentiating bacterial from viral infections in patients with sepsis.

## Figures and Tables

**Figure 1 viruses-17-00997-f001:**
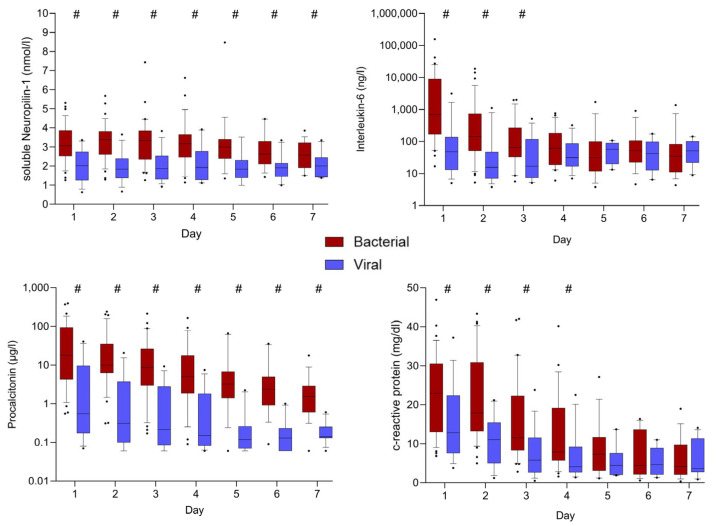
Comparison of the course of median inflammatory markers between bacterial and viral sepsis patients. #: indicates statistically significant differences between groups (*p* < 0.05).

**Table 1 viruses-17-00997-t001:** Comparison of the baseline characteristics between the bacterial and viral sepsis groups.

	Bacterial (n = 37)	Viral (n = 14)	*p*
Age °	66 (53–71)	65 (57–71)	0.849
Sex (male) *	26 (70.3%)	11 (78.6%)	0.553
BMI °	26.6 (24.7–30.6)	26.2 (23.2–28.4)	0.580
HbA1c% °	5.8 (5.6–6.4)	6.0 (5.9–6.5)	0.316
SOFA score °	9 (8–11)	7 (6–8)	0.001
SAPS III °	65 (61–76)	61 (46–65)	0.018
Vasopressor administration at admission *	34 (94.4%)	11 (78.5%)	0.140
Norepinephrin at admission (µg/kg/min) °	0.23 (0.10–0.37)	0.17 (0.08–0.21)	0.137
Vasopressin at admission (U/hr) °	1.6 (1.6–1.6)	0.8 (0.8–0.8)	0.182
**Inflammatory markers at ICU admission**			
Interleukin-6 (ng/L) °	3900.5 (984.0–44,140.0)	145.0 (68.4–190.0)	<0.001
C-reactive protein (mg/dL) °	17.8 (10.7–29.5)	10.6 (4.6–19.5)	0.080
Procalcitonin (µg/L) °	29.1 (5.5–63.8)	0.3 (0.2–0.6)	<0.001
**Comorbidities**			
Hypertension *	16 (43.2%)	7 (50.0%)	0.665
Coronary artery disease *	16 (43.2%)	6 (42.9%)	0.980
Atrial fibrillation *	9 (24.3%)	3 (21.4%)	0.828
COPD *	8 (21.6%)	1 (7.1%)	0.226
Diabetes mellitus type I *	1 (2.7%)	0	0.534
Diabetes mellitus type II *	7 (18.9%)	3 (21.4%)	0.840
Hepatic comorbidity *	7 (18.9%)	0	0.080
Pulmonary comorbidity *	3 (8.1%)	1 (7.1%)	0.909
Chronic kidney disease *	6 (16.2%)	3 (21.4%)	0.663
**Focus of infection**			
Respiratory *	17 (45.9%)	14 (100%)	0.014
Gastrointestinal *	6 (16.2%)	0	
Genitourinary *	4 (10.8%)	0	
Cutaneous *	7 (18.9%)	0	
Other *	3 (8.1%)	0	
**Hospital stay**			
Length of stay in ICU °	7 (3–19)	17 (7–25)	0.062
Length of stay in hospital °	17 (7–48)	24 (11–61)	0.299
Bacterial co-infection during stay *	15 (40.5%)	9 (64.3%)	0.129
Fungal co-infection during stay *	17 (45.9%)	9 (64.3%)	0.242
ICU mortality *	11 (29.7%)	0	0.021
Hospital mortality *	12 (32.4%)	1 (7.1%)	0.064

* n (%); ° median (IQR); BMI: Body mass index; HBA1c%: glycated hemoglobin; SOFA: sequential organ failure assessment; SAPS III: simplified acute physiology score; COPD: chronic obstructive pulmonary disease; ICU: intensive care unit.

**Table 2 viruses-17-00997-t002:** Comparison of sNRP-1 levels between patients with bacterial sepsis and patients with viral sepsis.

	Bacterial Cohort (n = 37)	Viral Cohort (n = 14)	*p*
Day 1 (nmol/L)	3.06 (2.57–3.68)	2.02 (1.26–2.75)	0.004
Day 2 (nmol/L)	3.38 (2.63–3.76)	1.84 (1.44–2.09)	<0.001
Day 3 (nmol/L)	3.34 (2.38–3.84)	1.87 (1.32–2.54)	<0.001
Day 4 (nmol/L)	3.16 (2.59–3.61)	1.93 (1.31–2.49)	0.025
Day 5 (nmol/L)	2.99 (2.39–3.40)	1.84 (1.44–2.29)	0.009
Day 6 (nmol/L)	2.64 (2.11–3.29)	1.91 (1.50–2.10)	0.009
Day 7 (nmol/L)	2.58 (1.92–3.22)	2.00 (1.44–2.37)	0.040

**Table 3 viruses-17-00997-t003:** Area under the ROC curve (AUC) of sNRP-1 for discrimination between bacterial and viral sepsis.

	AUC (95% CI)
Day 1	0.777 (0.630–0.924)
Day 2	0.846 (0.718–0.974)
Day 3	0.823 (0.686–0.960)
Day 4	0.744 (0.545–0.943)
Day 5	0.807 (0.617–0.997)
Day 6	0.800 (0.619–0.981)
Day 7	0.739 (0.540–0.938)

**Table 4 viruses-17-00997-t004:** Comparison of sNRP-1 levels between patients with bacterial sepsis and patients with viral sepsis, restricted to patients with a respiratory focus.

	Bacterial Cohort (n = 17)	Viral Cohort (n = 14)	
Day 1 (nmol/L)	2.75 (2.48–3.43)	2.02 (1.26–2.75)	0.043
Day 2 (nmol/L)	3.25 (2.63–3.49)	1.84 (1.44–2.09)	0.001
Day 3 (nmol/L)	3.00 (2.25–3.76)	1.87 (1.32–2.54)	0.004
Day 4 (nmol/L)	3.04 (2.62–3.50)	1.93 (1.31–2.49)	0.030
Day 5 (nmol/L)	2.93 (2.55–3.47)	1.84 (1.44–2.29)	0.016
Day 6 (nmol/L)	2.61 (2.55–3.29)	1.91 (1.50–2.10)	0.016
Day 7 (nmol/L)	2.56 (1.91–3.03)	2.00 (1.44–2.37)	0.099

**Table 5 viruses-17-00997-t005:** Area under the ROC curve (AUC) of sNRP-1 for discrimination between bacterial and viral sepsis restricted to patients with a respiratory focus.

	AUC (95% CI)
Day 1	0.725 (0.533–0.918)
Day 2	0.837 (0.677–0.996)
Day 3	0.810 (0.645–0.976)
Day 4	0.769 (0.556–0.982)
Day 5	0.818 (0.608–1.000)
Day 6	0.809 (0.605–1.000)
Day 7	0.718 (0.492–0.944)

## Data Availability

The datasets used and/or analyzed during the current study are available from the corresponding author on reasonable request.

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
