# Peer review of "Soluble Neuropilin-1 as a Marker for Distinguishing Bacterial and Viral Sepsis in Critically Ill Patients—A Prospective, Multicenter, Observational Study"

_viruses, 2025, doi:10.3390/v17070997_

Round 1

Reviewer 1 Report

Comments and Suggestions for Authors

please see the attached file.

Comments on the Quality of English Language

Some grammar mistakes should be revised.

Reviewer 2 Report

Comments and Suggestions for Authors

The authors investigated whether soluble neuropilin-1 (sNRP-1) is suitable for differentiating between bacterial and viral causes of septic patients. After a brief introduction to the very relevant topic, the material and methods are presented; the patients were part of the SEPSIS-3 study. Please indicate the ethics vote. PCT, IL-6, CRP and sNRP-1 were determined in arterial plasma after storage at -80°C. A total of 51 patients could be included. IL-6 and CRP were only higher in the first 4 days of a bacterial infection than in viral infections, while PCT and sNRP-1 differed throughout the entire week of observation. From day 4 onwards, this was no longer significant for sNRP-1. The conclusion that this is a promising marker is probably a little too optimistic, but this may change with further findings on sNRP-1 regulation.
